# PRISM: Diversifying Dataset Distillation by Decoupling Architectural Priors

## Abstract

Dataset distillation (DD) promises compact yet faithful synthetic data, but existing approaches often inherit the inductive bias of a single teacher model. As dataset size increases, this bias drives generation toward overly smooth, homogeneous samples, reducing intra-class diversity and limiting generalization. We present PRISM (PRIors from diverse Source Models), a framework that disentangles architectural priors during synthesis. PRISM decouples the logit-matching and regularization objectives, supervising them with different teacher architectures: a primary model for logits and a stochastic subset for batch-normalization (BN) alignment. On ImageNet-1K, PRISM consistently outperforms single-teacher methods (e.g., SRe2L) and recent multi-teacher variants (e.g., G-VBSM) at low- and mid-IPC regimes. The generated data also show significantly richer intra-class diversity, as reflected by a notable drop in cosine similarity between features. We further analyze teacher selection strategies (pre- vs. intra-distillation) and introduce a scalable cross-class batch formation scheme for fast parallel synthesis. Code will be released after the review period.

## 1 Introduction

Dataset distillation (DD) has emerged as a critical controllable data generation method in modern deep learning, motivated by three core objectives: enhanced robustness against adversarial attacks (Lai et al., 2025), improved privacy through membership and model inversion safeguards (Dong et al., 2022; Carlini et al., 2022), and efficient data compression (Zhao et al., 2020; Zhao & Bilen, 2023). Unlike large generative visual models such as diffusion models, DD produces images whose class semantics are guaranteed by gradient supervision, i.e., they produce training-signal-equivalent samples (Dhariwal & Nichol, 2021; Fort & Whitaker, 2025; Cazenavette et al., 2022).

As DD techniques have matured, a significant challenge remains unresolved: the failure to synthesize diverse, large-scale datasets that capture the complexity of their real-world counterparts. Current DD methods, including gradient matching (Yin et al., 2023) or parameter matching (Cazenavette et al., 2022; Cui et al., 2023), also in combination with generative priors (Cazenavette et al., 2023; Moser et al., 2024; Su et al., 2024; Duan et al., 2023), predominantly concentrate on high-resolution synthetic images in small-scale settings, such as 50 or 100 images-per-class (IPC) and fail to close the gap between the full and compressed datasets.

In this work, we question the motivation of compressing a dataset, since *memory is cheap*, in favor of the other motivations, namely robustness and privacy. Yet, naively upscaling DD techniques for an identical dataset size often results in overly smooth, feature-restricted datasets lacking sufficient diversity (Shao et al., 2024a; Sun et al., 2024; Shen et al., 2025), as illustrated in Figure 1. This leads to *homogeneous representations devoid of sufficient intra-class variability*, thus impairing robustness as well as safety and ultimately limiting the practical utility of DD (Sorscher et al., 2022).

Our central argument is that knowledge within a trained network is inseparable from the architecture that contains it (Ulyanov et al., 2018; Shao et al., 2024a). Any single model possesses a strong inductive bias, i.e., its "view" of the world. Distilling a dataset through one model inevitably imprints this single, limited view onto the synthetic data, resulting in a homogenous dataset that fails to train generalizing models. *In order to create a truly generalizable synthetic dataset, we must synthesize it from a distribution of "world views"*.

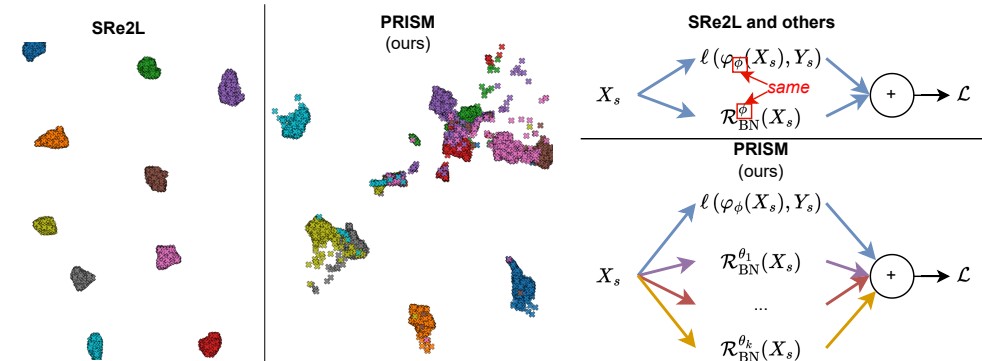

Figure 1: UMAP visualization of synthetic images from ImageNet-1K (10 classes), comparing SRe2L with our proposed multi-teacher alignment. Our approach, PRISM, generates significantly greater intra-class diversity, contrasting the overly uniform clusters of SRe2L that can lead to model overfitting more easily.

Figure 2: The core idea behind PRISM (**PRI**ors from diverse **S**ource **M**odels): Use multiple, diverse models for decoupling the logit maximization and regularization through BN alignment instead of one, like in SRe2L and related work.

Addressing this limitation, we propose diversifying the distillation process through a multi-architectural prior framework that simultaneously optimizes logit matching (Yin et al., 2023) and regularization through batch normalization (BN) alignment (Yin et al., 2020) with at least two decoupled teacher models, which we coin PRISM (**PRI**ors from diverse **S**ource **M**odels). As such, our teacher decoupling is orthogonal to all existing scaling attempts, as shown in Figure 2. Instead of relying on training schedules (Shen et al., 2025), data initialization (Sun et al., 2024), or post-evaluation pipelines (Shao et al., 2024b), PRISM introduces an orthogonal mechanism by **decoupling the logit-matching objective from the BN-alignment regularization**. This allows different architectural priors - from multiple, distinct teacher models - to simultaneously contribute to different aspects of the synthesis, complementing rather than replacing prior innovations.

In summary, our main contributions are threefold:

- First, we introduce PRISM, a novel framework that tackles the lack of diversity in dataset distillation by decoupling the architectural priors used for logit supervision and BN-alignment regularization.

- Second, we provide a systematic analysis of teacher-selection strategies, demonstrating that a pre-distillation selection of diverse teachers is highly effective.

- Third, we show that PRISM not only sets new state-of-the-art results, achieving up to **70.4%** top-1 accuracy with a ResNet-101 at IPC=100, but also generates datasets with quantifiably greater intra-class diversity, directly addressing the critical challenge of homogeneity in modern dataset distillation, while still maintaining a simple and massively parallelizable synthesis pipeline that scales efficiently to large datasets like ImageNet-1K.

## 2 RELATED WORK

**Dataset Distillation.** The field has largely bifurcated into two main strategies. Early and prominent methods relied on **batch-to-batch matching**, which involves expensive bi-level optimization to align gradients (Zhao et al., 2020), training trajectories (Cazenavette et al., 2022; Cui et al., 2023), or feature distributions (Zhao & Bilen, 2023) between real and synthetic data batches. While effective, the computational overhead of these methods makes them challenging to apply to large-scale datasets.

Thus, the community has shifted towards **batch-to-global matching**, a paradigm pioneered by SRe2L (Yin et al., 2023). This approach uses a pre-trained teacher model to generate global statistics (e.g., from batch normalization layers) and then optimizes synthetic images to match these global targets. This strategy is significantly more efficient and has enabled distillation at the scale of ImageNet-1K.

However, because every synthetic image in a class is optimized against the same global signal, this approach often suffers from a lack of intra-class diversity (Shao et al., 2024a).

**Strategies for Enhancing Diversity.** Recognizing the diversity challenge in batch-to-global matching, several methods have proposed alternative solutions. **G-VBSM** (Shao et al., 2024a) was a key step in this direction, introducing the idea of using multiple teacher models and statistical metrics (beyond just BN statistics) to create a richer, more varied supervision signal. **EDC** (Shao et al., 2024b) further refined this multi-teacher approach by identifying a suite of critical, and often overlooked, design choices in the post-training and evaluation pipeline, such as smoothing learning rate schedules and EMA-based evaluation, that are necessary to unlock the full potential of a diverse distilled dataset.

Other methods have approached diversity from different angles. **RDED** (Sun et al., 2024) focuses on the initialization data itself, using multi-crop image concatenation to create varied starting points for distillation. While this improves performance and speed, it sidesteps the goal of generating purely synthetic data, as the resulting images are composites of real images, which may not satisfy the privacy and robustness motivations of DD. In contrast, **DELT** (Shen et al., 2025) introduces the "EarlyLate" training scheme, where images are synthesized for varying numbers of iterations to create a spectrum of synthetic samples - from realistic to abstract.

## 3 METHODOLOGY

In the following, we will derive the method behind PRISM step-by-step, starting from classical DD and SRe2L. Consider a real dataset $\mathcal{T} = (X_r, Y_r)$ comprising $N$ images, where $X_r \in \mathbb{R}^{N \times H \times W \times C}$ are the real images. DD aims to distill this dataset into a smaller synthetic set $\mathcal{S} = (X_s, Y_s)$, where $X_s \in \mathbb{R}^{M \times H \times W \times C}$ with $M \ll N$. Conventionally, $M = \mathcal{C} \cdot \text{IPC}$, where $\mathcal{C}$ is the number of classes and IPC are the specified images-per-class. Formally, classical DD seeks the optimal synthetic dataset $\mathcal{S}^*$ that minimizes the distillation loss $\mathcal{L}(\mathcal{S}, \mathcal{T})$:

$$\mathcal{S}^* = \arg \min_{\mathcal{S}} \mathcal{L}(\mathcal{S}, \mathcal{T}). \tag{1}$$

Here, we follow the formulation of SRe2L (Yin et al., 2023) by optimizing over an output of a teacher model $\varphi_{\boldsymbol{\phi}}$ with parameters $\boldsymbol{\phi}$:

$$X_S^* = \arg \min_{X_S} \ell \left( \varphi_{\boldsymbol{\phi}}(X_s), \boldsymbol{Y_s} \right) + \lambda \mathcal{R}_{\text{reg}}, \tag{2}$$

where $\mathcal{R}_{\text{reg}}$ is the regularization term to avoid noise-like artifacts, i.e., regularize synthetic images to look more natural, and $\lambda$ is its weighting hyperparameter. While there are multiple valiable options for $\mathcal{R}_{\text{reg}}$, such as L2 or TV regularization, the authors of SRe2L found that using the deep inversion (Yin et al., 2020) inspired BN alignment $\mathcal{R}^{\boldsymbol{\theta}}_{\text{BN}}$ of the model alone led to the best overall performance:

$$\mathcal{R}_{\text{reg}} = \mathcal{R}^{\boldsymbol{\theta}}_{\text{BN}}(X_s) = \sum_l \left\| \mu_{l,\boldsymbol{\theta}}(X_s) - \mathbb{E} \left( \mu_{l,\boldsymbol{\theta}} \mid \mathcal{T} \right) \right\|_2 + \sum_l \left\| \sigma^2_{l,\boldsymbol{\theta}}(X_s) - \mathbb{E} \left( \sigma^2_{l,\boldsymbol{\theta}} \mid \mathcal{T} \right) \right\|_2, \tag{3}$$

where $l$ is the index of BN layer in the model with parameters $\theta$, $\mu_{l,\boldsymbol{\theta}}(\widetilde{\boldsymbol{x}})$ and $\sigma^2_{l,\boldsymbol{\theta}}(\widetilde{\boldsymbol{x}})$ are mean and variance, which can be conveniently approximated by the running mean and running variance in a pre-trained model at the $l$-th layer.

### 3.1 DUAL-TEACHER DECOUPLING

The standard SRe2L framework (Yin et al., 2023) uses a single, pre-trained teacher model for both parts of the objective function. This means the same model architecture and weights provide the supervision for both the logit-matching term (governed by parameters $\boldsymbol{\phi}$) and the BN-alignment regularization (which depends on the BN layer parameters within $\boldsymbol{\theta}$). We refer to this standard approach, where a single model fulfills both roles, as **single-teacher alignment**. Thus, $\boldsymbol{\phi} = \boldsymbol{\theta}$.

The core idea of PRISM is to challenge this coupling. We propose to **decouple** the architectural priors by allowing different models to supervise each term, a strategy we coin **multi-teacher alignment**. In this setup, the logit teacher's parameters ($\boldsymbol{\phi}$) and the BN teacher's parameters ($\boldsymbol{\theta}$) belong to distinct models. For instance, one could use an EfficientNet as the logit teacher and a standard ResNet as

the BN teacher. This leads to an optimization where the gradient is a composite of two different architectural perspectives:

$$\nabla_{X_s}\mathcal{L}(\mathcal{S},\mathcal{T}) = \underbrace{\nabla_{X_s}\ell(\varphi_\phi(X_s), Y_s)}_{\text{Teacher 1}} + \lambda\underbrace{\nabla_{X_s}\mathcal{R}_{BN}^\theta(X_s)}_{\text{Teacher 2}}$$

As a result, the optimization of $X_s$ is guided by two distinct and potentially complementary objectives derived from different architectural priors:

- $\nabla_{X_s}\ell(\varphi_\phi(...))$: This term pushes $X_s$ to have features that are effective for classification from the perspective of the logit teacher. If optimized in isolation, this objective is known to produce adversarial-like patterns that lack semantic realism and, therefore, lead to poor generalization (Yin et al., 2020).
- $\nabla_{X_s}\mathcal{R}_{BN}^\theta(...)$: This term pushes $X_s$ to have low-level global feature statistics (mean and variance) that are considered "natural" from the perspective of the BN teacher model with parameters $\theta$, a countermeasure against adversarial-like patterns.

### 3.2 GENERALIZED MULTI-TEACHER ALIGNMENT

As a natural next step, a more generalized version of our proposed decoupling is to apply multiple models for BN alignment. Thus, we obtain **PRI**ors from diverse **S**ource **M**odels (**PRISM**). More concretely, let $\mathcal{M} = \{\varphi_{\boldsymbol{\theta}_1}, \ldots, \varphi_{\boldsymbol{\theta}_k}\}$ be $k$ models used for BN alignment, the objective becomes:

$$X_S^* = \underset{X_S}{\arg\min}\, \ell\left(\varphi_\phi(X_s), \boldsymbol{Y}_s\right) + \lambda\mathcal{R}_{\text{reg}}$$

$$= \underset{X_S}{\arg\min}\, \ell\left(\varphi_\phi(X_s), \boldsymbol{Y}_s\right) + \lambda\sum_{\boldsymbol{\omega}\in\mathcal{M}}\mathcal{R}_{\text{BN}}^{\boldsymbol{\omega}}(X_s) \tag{4}$$

This core principle is visualized in Figure 2. To further boost diversity, we recommend including probabilities of using more than one BN alignment for each distilled image. To formalize this, let $\mathcal{M}_{\text{pool}} = \{\varphi_{\boldsymbol{\theta}_1}, \ldots, \varphi_{\boldsymbol{\theta}_{k_{\text{total}}}}\}$ be the full set of $k_{\text{total}}$ available source models for BN alignment, and let $k_{\text{max}}$ be the maximum number of BN teachers to use simultaneously, constrained by VRAM. We define the set of all valid, non-empty subsets of teachers as $\mathbb{M}_{\text{valid}} = \{\mathcal{M}_{\text{sub}} \subseteq \mathcal{M}_{\text{pool}} \mid 1 \leq |\mathcal{M}_{\text{sub}}| \leq k_{\text{max}}\}$.

Thus, we sample a subset $\mathcal{M}_{\text{sub}}$ from a distribution $P$ over all possible valid subsets, $\mathcal{M}_{\text{sub}} \sim P(\mathbb{M}_{\text{valid}})$. A simple and effective choice for $P$ is the uniform distribution. The overall objective is then to minimize the expected loss over this random selection:

$$X_S^* = \underset{X_S}{\arg\min}\, \ell\left(\varphi_\phi(X_s), \boldsymbol{Y}_s\right) + \mathbb{E}_{\mathcal{M}_{\text{sub}}\sim P(\mathbb{M}_{\text{valid}})}\left[\lambda\sum_{\boldsymbol{\omega}\in\mathcal{M}_{\text{sub}}}\mathcal{R}_{\text{BN}}^{\boldsymbol{\omega}}(X_s)\right] \tag{5}$$

By following the insights of Tran et al. (2021), we further claim improved robustness by this generalized multi-teacher alignment by providing a proof sketch in the appendix.

### 3.3 TEACHER-SELECTION STRATEGY

To further dissect the role of teacher model selection in distillation, we explore two distinct teacher-selection strategies: (1) a **pre-distillation selection** strategy, in which a fixed set of teacher models is determined before the distillation begins, and (2) an **intra-distillation selection** strategy, where teachers are dynamically selected during the distillation process itself.

More specifically, in the **pre-distillation** strategy, the set of active teachers is determined once for each synthetic image before the optimization process begins. For a given image $X_s$, we sample a single gradient teacher $\varphi_\phi$ and a corresponding subset of BN alignment teachers $\mathcal{M}_{\text{sub}}$. This fixed ensemble then guides the entire distillation process for that specific image.

Conversely, the **intra-distillation** strategy leads to a more dynamic distillation process by re-selecting teachers *during* the optimization itself, a method heavily inspired by G-VBSM (Shao et al., 2024a). Within PRISM, this means that at each distillation step, a new set of teachers, both the gradient teacher $\varphi_\phi$ and the BN alignment subset $\mathcal{M}_{\text{sub}}$, can be re-sampled from their respective pools.

Table 1: Comparison with state-of-the-art dataset distillation on **ImageNet-1K**. We report top-1 accuracy (%) for ResNet-18/50/101 trained on distilled datasets at IPC = 10, 50, 100 reporting mean±std over three seeds. While competitive at low IPCs, our method, PRISM, consistently establishes new state-of-the-art performance at higher IPCs (50 and 100) across all architectures and evaluation protocols. Results marked with ‡ use DELT's evaluation procedure (Shen et al., 2025). Bold indicates the best value per column. All other results follow our validation protocol. "–" denotes not reported.

| Method | ResNet-18 | | | ResNet-50 | | | ResNet-101 | | |
|---|---|---|---|---|---|---|---|---|---|
| | (IPC=10) | (IPC=50) | (IPC=100) | (IPC=10) | (IPC=50) | (IPC=100) | (IPC=10) | (IPC=50) | (IPC=100) |
| SRe2L | 21.3±0.6 | 46.8±0.2 | 52.8±0.3 | 28.4±0.1 | 55.6±0.3 | 61.0±0.4 | 30.9±0.1 | 60.8±0.5 | 62.8±0.2 |
| G-VBSM | 31.4±0.5 | 51.8±0.4 | 55.7±0.4 | 35.4±0.8 | 58.7±0.3 | 62.2±0.3 | 38.2±0.4 | 61.0±0.4 | 63.7±0.2 |
| RDED | 42.0±0.1 | 56.5±0.1 | 59.8±0.1 | - | - | - | 30.9±0.1 | 60.8±0.5 | 62.8±0.2 |
| EDC | 48.6±0.3 | 58.0±0.2 | - | **54.1±0.2** | 64.3±0.2 | - | **51.7±0.3** | 64.9±0.2 | - |
| **PRISM** | **49.4±0.2** | **59.0±0.1** | **60.9±0.2** | 51.1±1.2 | **65.1±0.1** | **67.5±0.2** | 48.5±1.7 | **65.9±0.2** | **68.6±0.4** |
| DELT ‡ | 46.1±0.4 | 59.2±0.4 | 62.4±0.2 | - | - | - | **48.5±1.6** | 66.1±0.5 | 67.6±0.3 |
| **PRISM ‡** | **46.9±0.1** | **59.6±0.2** | **62.7±0.1** | **46.3±0.7** | **66.5±0.2** | **69.4±0.1** | 40.7±3.6 | **66.7±0.2** | **70.4±0.2** |

## 3.4 BATCH FORMATION AND PARALLELIZATION STRATEGY

A key design choice that distinguishes PRISM from other recent methods like G-VBSM (Shao et al., 2024a), EDC (Shao et al., 2024b), and DELT (Shen et al., 2025) is the batch formation strategy during data synthesis. PRISM, following SRe2L, processes each image-per-class (IPC) index independently. This creates *cross-class* batches where each batch consists of a single IPC slice across multiple classes (e.g., the $i$-th image from each class). The primary advantage of this strategy is its high efficiency and straightforward parallelizability; the synthesis of each IPC can be treated independently and easily distributed across multiple GPUs.

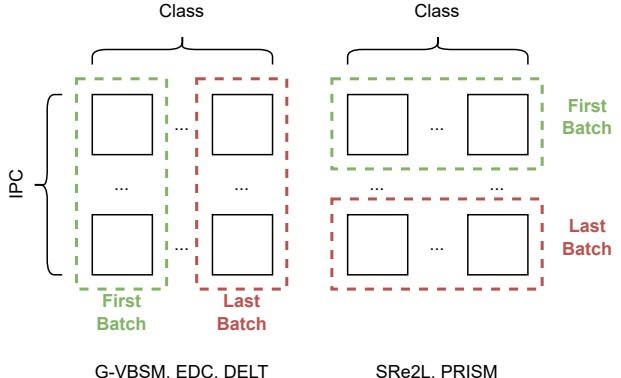

Figure 3: Batch formation and optimization strategies. **(Left)** Methods like G-VBSM, EDC, and DELT optimize jointly over all classes simultaneously. **(Right)** Methods like our PRISM and SRe2L, process each IPC index independently.

In contrast, other approaches often form *intra-class* batches, which contain multiple images from the same class, as shown in Figure 3. This enables specific regularization, such as the data densification in G-VBSM/EDC or diversity-driven optimization in DELT, which operates on the same-class images within a batch. While this improves diversity, this also comes at the cost of increased complexity, as such regularizations introduce intra-batch dependencies during optimization, e.g., through explicitly pushing images during distillation apart. Our method prioritizes a simple and massively parallelizable pipeline, achieving diversity not through complex intra-batch information exchange between images, but through our core contribution of BN decoupling and diversifying architectural priors.

## 4 EXPERIMENTS

### 4.1 SETUP

We used the SRe2L pipeline (Yin et al., 2023) and its standard configurations for the recovery stage. Moreover, we initialized the synthetic dataset by selecting each real image exactly once from the ImageNet training set, ensuring a direct and consistent comparison across different configurations in our identical dataset-size distillation (no coreset selection like in DELT or multi-image initialization as in RDED/EDC). During the distillation process, we evaluated multiple teacher models to diversify synthetic data representations. We set a maximum of 4000 optimization iterations per image.

Table 2: Study of **teacher alignment and selection** strategies on **ImageNet-1K**, **IPC≈1200**: final validation accuracy [%] of **ResNet-18**. In addition, we report the max. VRAM consumption during distillation under a distillation batch size of 100. Note that this is the result for recovery-only (no knowledge distillation).

| Variant | Model Selection | BN Teachers | Acc. [%] | VRAM [GB] |
|---|---|---|---|---|
| Baseline (real data) | - | - | 70.0 | - |
| SRe2L | - | 1 | 17.9 | 6.5 |
| + single-teacher alignment | intra-distillation | 1 | 18.3 | 6.5 |
| + dual-teacher decoupling | intra-distillation | 1 | 19.0 | 13.0 |
| + single-teacher alignment | pre-distillation | 1 | 32.4 | 6.5 |
| **+ dual-teacher decoupling** | **pre-distillation** | **1** | **36.2** | **13.0** |
| + multi-teacher alignment | pre-distillation | 2 | 37.4 | 18.5 |
| + multi-teacher alignment | pre-distillation | 3 | 38.7 | 26.0 |
| **+ multi-teacher alignment** | **pre-distillation** | **4** | **39.1** | **32.5** |

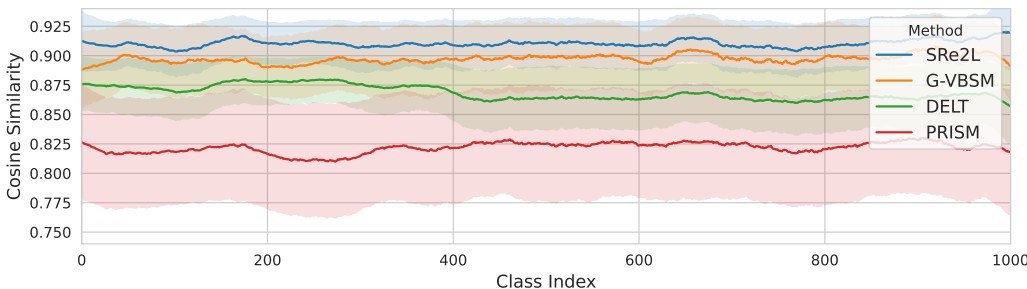

Figure 4: **Intra-class semantic cosine similarity** with a pretrained ResNet-18 model on ImageNet-1K dataset applied on the respective distilled images showing higher diversity as indicated by lower mean values and higher variance.

## 4.2 CLASSICAL DATASET DISTILLATION

Table 1 summarizes our main results, comparing PRISM against state-of-the-art methods on ImageNet-1K. To ensure a fair and comprehensive analysis, we present results under two distinct evaluation protocols: our own, which is optimized for low-to-mid IPCs (more details follow in Section 4.4), and the protocol used by DELT, which excels at higher IPCs.

**Performance with PRISM's Optimized Evaluation.** Under our primary evaluation protocol, PRISM establishes new SOTA results compared to EDC in Table 1. While EDC shows strong performance at IPC=10 on larger backbones, PRISM consistently outperforms all prior methods at IPC=50 and IPC=100 across all tested architectures. For instance, on ResNet-18, PRISM achieves an accuracy of **49.4% at IPC=10**, surpassing EDC (48.6%), and extends its lead at **IPC=50 (59.0%)** and **IPC=100 (60.9%)**. This trend holds for larger models, where PRISM's performance of **65.1% (ResNet-50, IPC=50)** and **68.6% (ResNet-101, IPC=100)** confirms that decoupling architectural priors is a highly effective strategy for generating diverse and generalizable synthetic data.

**Performance with DELT's Evaluation Procedure.** When evaluated using the DELT protocol, PRISM's advantages become even more pronounced for mid-to-high IPC scenarios, setting new SOTA results across multiple settings in Table 1. For ResNet-18, PRISM outperforms DELT at **IPC=50 (59.6%)** and **IPC=100 (62.7%)**. The benefits of our diverse architectural priors scale to larger models, where PRISM achieves a remarkable **69.4% accuracy on ResNet-50** and **70.4% on ResNet-101** at IPC=100. PRISM's ability to excel under an evaluation pipeline tailored for a different method underscores the fundamental quality of the dataset it generates, proving its robustness across varied training configurations, especially for larger IPCs.

SRe2L
PRISM

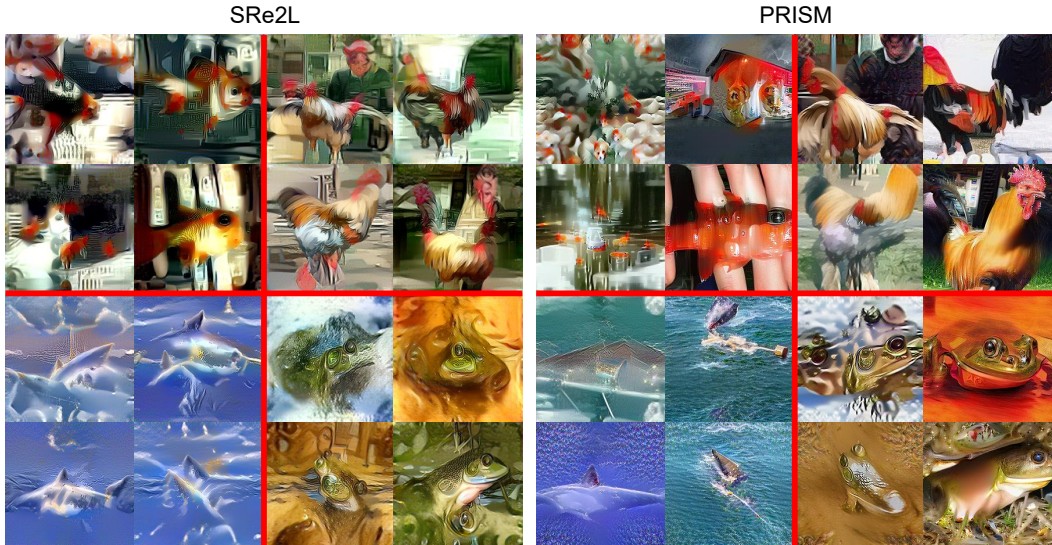

Figure 5: **Qualitative comparison of synthetic images** from ImageNet-1K generated by SRe2L and PRISM. Both methods start from the *exact same* initial real images to ensure a fair comparison. The images generated by SRe2L **(left)** exhibit significant homogeneity, with samples within each class (goldfish, rooster, shark, frog) converging to similar colors and textures. In contrast, PRISM **(right)** produces a wider variety of contexts and colorations.

### 4.3 ANALYSIS ON RECOVERY-ONLY DIVERSITY

Recent advances in dataset distillation have increasingly coupled the core data synthesis process with powerful post-recovery optimizations, such as knowledge distillation (Qin et al., 2024; Li et al., 2025) and highly specialized validation schedules (Shao et al., 2024b; Shen et al., 2025; Yin & Shen, 2023). While effective at boosting final accuracy, this entanglement can obscure the intrinsic quality of the generated data, making it difficult to attribute performance gains directly to the synthesis method itself. Therefore, to perform a rigorous and unbiased assessment of our architectural decoupling, this analysis intentionally isolates the synthesis stage from these downstream optimizations. Our rationale is that a truly effective diversity-enhancing method like PRISM should demonstrate a clear advantage in this controlled setting, directly linking its architectural design to the quality of the distilled data. Experimental details for this setup without knowledge distillation are outlined in in the appendix.

Table 2 summarizes the results of our recovery-only setting, from which we can derive the following three positive observations: **(i)** Sampling multiple teacher models during distillation helps to improve the performance, but pre-selecting them for the whole distillation process is better. **(ii)** Decoupling gradient matching and BN statistics alignment further improves the performance. **(iii)** The best performance is achieved by using multiple models for BN statistics alignment. Taken together, all best performing configurations form our proposed method PRISM, namely multi-teacher alignment with 4BN priors and pre-distillation selection.

To quantitatively validate that our method generates a more diverse dataset, we computed the intra-class semantic cosine similarity. This metric measures the average feature similarity between all pairs of synthesized images within the same class, using a pretrained ResNet-18 as a feature extractor; a lower similarity score thus indicates higher intra-class diversity. Figure 4 shows a clear separation between the methods. While existing approaches like SRe2L, G-VBSM, and DELT produce highly similar images (between 0.86 and 0.92), PRISM consistently achieves the lowest cosine similarity across all classes by a significant margin (mean values of 0.83 and below). This result provides strong quantitative evidence for our central claim: by decoupling and diversifying architectural priors, PRISM effectively breaks the homogeneity constraint inherent in single-teacher distillation, leading to the superior downstream performance reported in Table 1.

Table 3: Study of **relabeling** and additional **recovery** strategies on **ImageNet-1K** with **ResNet-18**, **IPC=10**: final validation accuracy [%].

| (a) **Soft-Label Targets** | | (b) **Add. Recovery Strategies** | | (c) **Batch Size (Relabeling)** | |
|---|---|---|---|---|---|
| Variant | Acc. [%] | Variant | Acc. [%] | Batch Size | Acc. [%] |
| ResNet-18 Only | 21.22 | Baseline from (a) | 29.53 | 1024 from (b) | 31.30 |
| Ensemble Average | 23.75 | + Resize Schedule | 29.93 | 16 | 40.41 |
| + MAE (GT=0.05) | 28.97 | **+ Var. Iterations** | **31.30** | 32 | 43.40 |
| + MSE (GT=0.05) | 28.07 | + Both | 30.89 | **50** | **44.04** |
| + MAE (GT=0.1) | 23.85 | | | 64 | 43.79 |
| **+ MSE (GT=0.1)** | **29.53** | | | 128 | 42.59 |

| (d) **Teacher Strategies** | | (e) **LR (Recovery)** | | (f) **Backbones** | |
|---|---|---|---|---|---|
| Variant | Acc. [%] | LR | Acc. [%] | Model Pool | Acc. [%] |
| Baseline from (c) | 44.04 | 0.25 from (d) | 45.73 | Models from (e) | 47.35 |
| $\phi = \theta_1 \neq \theta_{>1}$ | **45.73** | 0.10 | 46.63 | - EfficientNet (Rel.) | 47.53 |
| | | **0.05** | **47.35** | + AlexNet (Rel.) | |
| | | | | **+ AlexNet (Rec.)** | **47.77** |

This quantitative evidence is further supported by a direct qualitative comparison, as shown in Figure 5. To ensure a fair assessment, both PRISM and SRe2L started from the exact same initial images. While SRe2L consistently produces homogeneous images where samples from the same class converge on similar textures and poses, PRISM generates a visibly more diverse set.

During this study, we also experimented with alternative regularization approaches besides BN alignment, which can be found in the appendix. In more detail, we tried multi-resolution synthesis as an alternative regularization to BN alignment with CLIP embeddings (see **??**) as well as using large-scale pre-trained generative text-to-image models for identical dataset-size generation (see **??**).

### 4.4 RECOVERY & POST-RECOVERY STRATEGIES

To identify the optimal configuration for PRISM besides DELT's evaluation procedure, we began with the vanilla settings of SRe2L and systematically evaluated a series of improvements across the relabeling and recovery stages. Our analysis started with the generation of soft labels, then progressed to the image recovery process, and concluded with teacher model alignment and learning rate schedule refinements. The architectures employed during our multi-teacher selection were ResNet18, ShuffleNetV2-0.5, MobileNetV2, and EfficientNet-B0, with ResNet18 used for logit maximization to ensure comparability with G-VBSM (Shao et al., 2024a).

**Optimizing Soft-Label Generation.** Our first step was to refine the soft-label targets used for distillation. We found that moving beyond a single relabeler to an **ensemble of the recovery models** significantly improved performance, an idea inspired by G-VBSM (Shao et al., 2024a). As shown in Table 3a, this ensemble approach was most effective when using a **Mean Squared Error (MSE) loss** combined with a **0.1 ground-truth (GT) addition**, which proved superior to both MAE and standard KL divergence.

**Refining the Recovery Process.** With improved soft labels, we turned our attention to the recovery stage itself. We investigated two key strategies: scheduling the augmentation strength and varying the number of distillation iterations. Surprisingly, while scheduling the minimum crop size of the random resize augmentation offered a minor improvement, a simplified strategy of **varying the distillation iterations per image**, inspired by DELT (Shen et al., 2025), yielded the best performance in our setup (Table 3b).

**Batch Size during Relabeling.**   With an effective loss function established, we next examined the batch size used during this relabeling phase. Our experiments, summarized in Table 3c, revealed that a batch size of **50 was optimal**, aligning with similar findings in recent literature (Yin & Shen, 2023; Shao et al., 2024b). This result underscores the sensitivity of the relabeling process to batch statistics, even when using a distilled dataset.

**First BN teacher.**   Finally, we addressed the composition of the teacher models themselves. Confirming a key insight from SRe2L (Yin et al., 2023), we found it crucial that the primary logit matching teacher ($\phi$) and the primary BN alignment teacher ($\theta_1$) align with the model used for relabeling. Fixing this alignment provided a notable performance increase (Table 3d).

**Learning Rate in Recovery.**   We then fine-tuned the recovery learning rate. While the process began with a standard learning rate of 0.25, our ablations demonstrated that a much lower learning rate led to more stable convergence and better final accuracy. As detailed in Table 3e, we identified an optimal learning rate of **0.05**.

**Backbone Models.**   Furthermore, we discovered that swapping the computationally heavier EfficientNet for a lightweight **AlexNet** in both the relabeling and recovery teacher pools provided an additional performance boost without compromising the architectural diversity (Table 3f).

**Learning Rate Schedule.**   The culminating improvement came from adopting the learning rate schedule proposed by EDC (Shao et al., 2024b). Applying the **SSRS decayed cosine schedule** with a slowdown coefficient of $\zeta = 2.5$ provided the final significant leap in performance, leading to our state-of-the-art results.

## 5   LIMITATIONS AND FUTURE WORK

While PRISM establishes a new, orthogonal axis for scaling dataset distillation, our work also opens several exciting avenues for future investigation. We outline the primary limitations of our current approach, each of which represents a promising direction for subsequent research.

**VRAM Constraints on Teacher Ensembles.**   PRISM's cross-class batching parallelizes cleanly across multiple GPUs, making the synthesis of individual IPCs highly efficient. However, the number of *simultaneous* BN teachers that can be used for a single image is ultimately constrained by VRAM. This presents a compelling opportunity to explore memory-efficient teacher ensembles, such as through model-offloading techniques or parameter-efficient fine-tuning.

**Reliance on Batch Normalization.**   Our current formulation leverages the rich statistical priors available in batch normalization layers, which are prevalent in many standard CNN architectures. Extending our decoupling framework to directly regularize using priors from models with alternative normalization schemes, such as LayerNorm or GroupNorm, represents a natural next step.

## 6   CONCLUSION

We addressed a critical bottleneck in dataset distillation: the tendency of single-teacher methods to generate homogeneous datasets with poor intra-class diversity due to a constrained inductive bias. We introduced PRISM, a simple yet powerful framework that diversifies the synthesis process by decoupling architectural priors. By separating the logit-matching objective from the batch normalization alignment and supervising each with different teacher models, PRISM effectively injects a richer, multi-faceted training signal into the synthetic data.

Our extensive experiments on ImageNet-1K validate this approach, demonstrating that PRISM not only achieves state-of-the-art performance but also produces datasets with quantifiably greater semantic diversity. Ultimately, PRISM establishes architectural decoupling as a new, orthogonal axis for scaling dataset distillation, paving the way for larger, more generalizable synthetic datasets for robust and privacy-preserving machine learning.

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
