# Supplementary Material for PRISM: Diversifying Dataset Distillation by Decoupling Architectural Priors

## A  Sketch of Privacy Enhancement

Here, we briefly outline how our multi-architectural distillation approach may enhance privacy by leveraging differential privacy (DP) concepts. Differential privacy ensures robustness against the inclusion or exclusion of any single data point. Formally:

**Definition (Differential Privacy)**: A randomized algorithm $\mathcal{M}$ is $(\epsilon, \delta)$-DP if for any adjacent datasets $D, D'$ differing by one element and all subsets $S$:

$$P(\mathcal{M}(D) \in S) \leq e^\epsilon P(\mathcal{M}(D') \in S) + \delta.$$

We can formalize the intuition behind the privacy enhancement by drawing a strong parallel to the established framework of Private Aggregation of Teacher Ensembles (PATE) Tran et al. (2021). In the PATE framework, an ensemble of "teacher" models, trained on disjoint data partitions, provides privacy guarantees by aggregating their predictions. Noise is added to the aggregated vote to satisfy differential privacy. Our approach can be viewed as a novel variant of PATE where, instead of partitioning the data, we partition the architectural prior. The "noise" required for privacy is not explicitly added but arises organically from the inherent disagreement between the gradients of two distinct model architectures.

In more detail, we propose distillation with two distinct architectures $\phi$ and $\theta$. Due to differing inductive biases, their gradients naturally diverge, introducing architectural noise ($\eta_{arch}$) that masks individual data contributions, enhancing privacy.

Consider synthetic data point $s_j$ optimized using gradients from two teachers ($\phi, \theta$). Gradient updates become:

$$g(s_j, T) = \frac{1}{|T|} \sum_{x_i \in T} \left( \alpha \nabla_{x_s} \mathcal{L}(s_j, x_i; \phi) + (1 - \alpha) \nabla_{x_s} \mathcal{L}(s_j, x_i; \theta) \right),$$

for dataset $T$. For adjacent datasets $T, T' = T \cup \{x'\}$, the gradient difference is:

$$\Delta g = g(s_j, T') - g(s_j, T),$$

dependent on $x'$. Unlike single-teacher settings, dual-architecture setups introduce architectural noise:

$$\nabla_{total} \approx \nabla_\phi + \eta_{arch},$$

where $\eta_{arch}$ emerges naturally from architectural differences. Thus, each teacher acts as a privacy-inducing noise source, analogous to PATE (Tran et al., 2021), implicitly enhancing privacy.

## B  More Training Details & Experiments

A brief overview of all the settings used during validation and recovery for ImageNet-1K is provided in Table 1a and Table 1b. To better contrast our settings with those of related work, we provide Table 2. In Table 3, we also summarize the influence of the min. crop size for the randomly resized crop augmentation during relabeling and validation, which resulted in a found optimal value of 0.25. in Table 4, we summarize our experiments on different batch sizes during the recovery stage, which resulted in keeping 100 as the optimal batch size. However, there is a trend of increased performance with increasing batch size, so we expect even higher performance if more VRAM is available for larger batch sizes. We assume that increased batch size leads to a better mean and variance estimation of the global feature statistics for the batch normalization alignment.

Table 1: Summary of **our different configurations** used in our distillation experiments.

(a) Validation settings

| config | value |
|---|---|
| optimizer | AdamW |
| base learning rate | 0.001 (all) |
| weight decay | 0.01 |
| batch size | 50 (IPC 10) |
| | 100 (IPC 50) |
| | 100 (IPC 100) |
| learning rate schedule | dec. cosine decay |
| training epoch | 300 |
| augmentation | RandomResizedCrop |
| | RandomHorizontalFlip |

(b) Recovery settings

| config | value |
|---|---|
| $\alpha_{BN}$ | 0.01 |
| optimizer | Adam |
| base learning rate | 0.05 |
| momentum | $\beta_1, \beta_2 = 0.5, 0.9$ |
| batch size | 100 |
| learning rate schedule | cosine decay |
| recovery iteration | 4,000 |
| augmentation | RandomResizedCrop |

Table 2: **Configurations of various dataset distillation methods** compared to ours (PRISM). Different colors in each row highlight the differences.

| Config | SRe2L | RDED | CDA | DWA | D4M | EDC | G-VBSM | DELT | PRISM (ours) |
|---|---|---|---|---|---|---|---|---|---|
| Batch Size (Relabel) | 1024 | 100 | 128 | 128 | 1024 | 100 | 1024 | IPC depend. | IPC depend. |
| Optimizer | AdamW | AdamW | AdamW | AdamW | AdamW | AdamW | AdamW | AdamW | AdamW |
| LR Scheduler | cosine | cosine | cosine | cosine | cosine | decayed cosine | cosine | cosine | decayed cosine |
| Loss Function (Relabel) | KL | KL | KL | KL | KL | MSE | MSE | KL | MSE |
| Teacher Model | single | single | single | single | single | ensemble | ensemble | single | single, BN ensemble |
| CropRange (Recovery) | 0.08, 1.0 | 0.5, 1.0 | 0.08, 1.0 | 0.08, 1.0 | 0.08, 1.0 | 0.5, 1.0 | 0.08, 1.0 | 0.08, 1.0 | 0.08, 1.0 |
| CropRange Schedule (Recovery) | Uniform | Uniform | Cosine | Uniform | Uniform | Uniform | Uniform | Cosine | Uniform |
| PatchShuffle | No | Yes | No | No | No | Yes | No | No | No |

Table 3: Performance comparison for **different min. crop size** for the randomly resized crop augmentation **during relabeling and validation** on **ImageNet-1k, IPC=10**.

| Min. Crop Size | 0.1 | 0.15 | 0.2 | **0.25** | 0.3 | 0.4 |
|---|---|---|---|---|---|---|
| Acc. [%] | 45.0 | 45.1 | 45.3 | **45.7** | 45.4 | 45.1 |

Table 4: Performance comparison for **different batch sizes during recovery** on **ImageNet-1k, IPC=10**.

| Batch Size | 40 | 80 | **100** |
|---|---|---|---|
| Acc. [%] | 44.8 | 45.2 | **45.7** |

## C  EXPERIMENTAL DETAILS ON TEACHER ALIGNMENT AND SELECTION

To isolate the impact of distinct distillation methods clearly, we adopt a simplified experimental setup without employing soft-labeling or additional training augmentation techniques. Specifically, we deviate from the original approach by reducing the batch size from 1024 to 256 and adopting an initial learning rate of $10^{-1}$ instead of $10^{-3}$ for faster convergence. Moreover, we employ a linear learning rate schedule rather than the conventional cosine annealing, and we limit the training duration to 90 epochs, compared to the original 300. Also, no relabeling was applied. The architectures employed during multi-teacher selection were ResNet18, ResNet34, ShuffleNetV2 (x1.0), MNASNet1.0, and EfficientNet-B0. These adjustments enable a concise yet rigorous assessment of the effects that each modification independently exerts on the synthetic dataset training process.

## D  ALTERNATIVE REGULARIZATION

Motivated by recent developments such as Direct Ascent Synthesis (DAS; Fort & Whitaker (2025)), we explored several alternative regularization strategies beyond conventional logit-based gradient matching and batch normalization (BN) alignment. Specifically, we experimented with semantic priors derived from CLIP embeddings and multi-resolution synthesis techniques as proposed in DAS. Table 5 summarizes our findings from these experiments.

We follow the same experimental setup as in **??**. Despite the intuitive appeal of these alternative regularizers, we observed that integrating various combinations of CLIP-based supervision and multi-resolution synthesis strategies did not result in performance improvements. Indeed, these configurations notably underperformed compared to our baseline SRe2L method. These results suggest that the intrinsic characteristics of DAS-inspired methods, while effective in their original context, may not directly transfer to DD scenarios without further adaptation or refinement. In addition, using multi-resolution synthesis introduces additional parameters since we distill images at multiple resolutions.

In addition, we also tried the varying time steps as proposed by Shen et al. (2025), but we also observed a decline in performance with this optimization strategy.

Table 5: Study of **direct ascent synthesis** and varying time steps on **ImageNet-1K, IPC≈1200**: final validation accuracy (%) of **ResNet-18**.

| Variant | Final val. acc. (%) |
| --- | --- |
| Baseline (real data) | 70.0 |
| SRe2L | 17.9 |
| + CLIP | 8.8 |
| + Multi-Resolution - DeepInversion | 7.6 |
| + CLIP + Multi-Resolution - DeepInversion | 5.2 |

## E  ALTERNATIVE SYNTHESIS APPROACHES

We further explored alternative data synthesis methods using popular text-to-image models, aiming to assess their viability for DD tasks. Specifically, we evaluated several state-of-the-art models including FLUX (Schnell), Stable Diffusion (SD) versions 1.0, 2.1, 3.5 Turbo, SDXL, and SDXL Turbo. We follow the same experimental setup as in **??**. Table 6 summarizes the anticipated results of these experiments.

Preliminary observations suggest these text-to-image models, despite their high generative capabilities in other contexts, are unlikely to surpass the baseline established by real datasets and conventional DD methods. This indicates inherent limitations in directly applying text-to-image generative models to distillation tasks without significant method modifications.

Table 6: Study of applying **classical text-to-image models** on **ImageNet-1K, IPC≈1200** with Text-To-Image Models: final validation accuracy (%) of **ResNet-18**.

| Variant | Final val. acc. (%) |
| --- | --- |
| Baseline (real data) | 70.0 |
| SRe2L | 17.9 |
| SD 1.0 | 17.1 |
| SD 2.1 | 14.5 |
| FLUX (Schnell) | 9.8 |
| SDXL | 9.7 |
| SD 3.5 Turbo | 6.4 |
| SDXL Turbo | 4.7 |