# OpenReview forum: "PRISM: Diversifying Dataset Distillation by Decoupling Architectural Priors"
_ICLR.cc/2026/Conference — ICLR 2026 Conference Withdrawn Submission_

### Official Review · Reviewer_iNou · 2025-10-24

**Soundness:** 2
**Presentation:** 2
**Contribution:** 2
**Rating:** 2
**Confidence:** 4

**Summary:**

This paper introduces PRISM, a dataset distillation framework designed to overcome the inductive bias from using a single teacher model. By decoupling logit matching and BN regularization across diverse teacher architectures, PRISM enhances intra-class diversity and improves generalization.

**Strengths:**

The presentation is clear and easy to follow.

**Weaknesses:**

The overall writing and experimental presentation are quite rough, and the manuscript is not ready for publication in its current form.

The proposed method shows very limited novelty, appearing to be only a marginal modification of G-VBSM without substantial conceptual advancement.

Several important results are missing from the main comparison tables. Even if the original paper did not report these results, the referenced baseline methods are open-sourced, and the authors are expected to run them to ensure a fair and complete evaluation.

Moreover, the paper lacks comparisons with recent methods in dataset distillation. The authors should include or discuss relevant up-to-date works such as:

	•	A Label is Worth a Thousand Images in Dataset Distillation
	•	Diversity-Driven Synthesis: Enhancing Dataset Distillation through Directed Weight Adjustment
	•	Dataset Distillation via the Wasserstein Metric

The ablation study is insufficient. In particular:

	•	Cross-architecture performance, which is crucial for evaluating ensemble-based approaches, has not been discussed.
	•	The effect of different teacher model combinations should be analyzed, as this is closely related to the contribution claim of “decoupling architectural priors.”

Additionally, it is unclear what the “??” in lines 407–408 refers to, which raises concerns about the completeness and clarity of the manuscript.

**Questions:**

pls see the Weaknesses

---

### Official Review · Reviewer_PrvM · 2025-10-26

**Soundness:** 2
**Presentation:** 2
**Contribution:** 2
**Rating:** 2
**Confidence:** 4

**Summary:**

This paper introduces PRISM, a dataset distillation framework that improves sample diversity by decoupling logit supervision and BN alignment across multiple teacher architectures. Experiments on ImageNet-1K show performance gains over existing methods such as SRe2L and DELT, especially at higher IPCs.

**Strengths:**

1. The method and experiments are presented in a structured and easy-to-follow manner.

2. From the experimental results, the proposed method is very effective.

**Weaknesses:**

1. The proposed idea is **highly similar to the CV-DD [1] paper** released on arXiv in early 2025, which also employs an ensemble-style multi-model framework combining multiple model predictions and BN distribution alignments. PRISM mainly removes the multi-prediction matching component from CV-DD [1] and retains only a single prediction alignment, making the modification relatively **minor and incremental**. Furthermore, the paper does not cite CV-DD [1], raising concerns about the authors’ awareness and acknowledgment of closely related prior work.

2. All experiments are conducted only on ImageNet-1K. Smaller-scale benchmarks such as Tiny-ImageNet or CIFAR-100 are missing, making it difficult to assess the general applicability of the method across dataset sizes.

3. Lack of Cross-Architecture Evaluation: The experiments are performed exclusively on ResNet architectures. There is no analysis on whether PRISM generalizes to other backbones such as EfficientNet, ViT, or MobileNet, which limits the claim of broader effectiveness.

4. Missing Theoretical Support: The paper provides no theoretical justification or analytical discussion explaining why decoupling priors or using multiple teachers should yield performance gains. This weakens the conceptual contribution and leaves the empirical improvements largely unexplained.

5. Code not provided, limiting reproducibility.

**Questions:**

1. Could the authors further clarify the differences between PRISM and CV-DD [1]? The two methods appear to be conceptually very similar, and the modification introduced in PRISM seems rather minor and incremental.

2. Since the authors remove the multi-prediction component from the original CV-DD [1] framework, I would like to understand the rationale behind this design choice. Is there any justification or theoretical reasoning for removing it? Have the authors conducted any experiments or analyses demonstrating that excluding multi-prediction alignment leads to the observed performance gain?

3. From Figure 1, the visualization of PRISM is very messy, different classes are mixed together, and I think Sre2l is better.

[1] Dataset Distillation via Committee Voting

---

### Official Review · Reviewer_Ghc3 · 2025-10-28

**Soundness:** 3
**Presentation:** 3
**Contribution:** 2
**Rating:** 4
**Confidence:** 5

**Summary:**

The paper tackles the lack of diversity in distilled data caused by the inductive bias of a single teacher architecture. The method decouples priors by a logit teacher for classification supervision, and multiple BN teachers for BN alignment. Experiments on ImageNet-1K show improvements over SRe2L, G-VBSM, EDC, and DELT, with notable gains at IPC=50 and 100.

**Strengths:**

- **Clear motivation**: The paper identifies an issue in prior DD works and presents a simple idea (multi-teacher decoupling) to mitigate it.
- **Technical soundness and orthogonality**: The decoupling formulation is well-grounded and orthogonal to existing improvements (e.g., DELT). The derivation is clear.
- **Clarity and completeness**: The manuscript is well-structured, and the supplementary material provides additional training configurations, privacy discussions, and ablation studies.

**Weaknesses:**

- **Lack of evidence for diversity claim**: The authors claim that PRISM produces distilled data with higher intra-class diversity and lower cosine similarity, but the evidence is limited to one similarity curve (Fig. 4) and qualitative samples. Besides, there is no comparison to the real dataset’s diversity level, leaving it unclear whether PRISM truly approximates or exceeds the diversity of raw data. The claim of “diverse distilled samples benefit to DD” remains unsubstantiated.
- **Marginal performance gains**: Although the paper reports SOTA numbers on ImageNet-1K, the improvements over the strong DELT baseline are mostly within 0.5–1.0%, which make it hard to attribute the improvement confidently to the proposed architectural decoupling. BTW, the results of resnet101@IPC=10 is significantly lower than the DELT and there is no explanation.
- **Limited evaluation scope**: Experiments are conducted solely on ImageNet-1K. The paper does not include results on smaller datasets or challenging subsets (e.g., Tiny Imagenet, ImageNetIDC, nette, or woof), nor any cross-domain evaluations (CIFAR). This lack of evaluation datasets limits the demonstrated generalization ability of PRISM.
- **CNN-only evaluation**: All experiments use only CNN-based architectures (ResNet, MobileNet, ShuffleNet, EfficientNet).
Modern architectures such as Vision Transformers (ViT) or ConvNeXt are missing, which weakens the central claim that PRISM generalizes across diverse inductive biases. Besides, the paper motivates PRISM as a way to “decouple architectural priors,” yet it does not evaluate whether synthetic data distilled by PRISM transfers well to various networks beyond resnet family (**No cross-architecture generalization analysis**).
- **Missing comparisons with modern generative DD methods**: The evaluation excludes diffusion-based dataset distillation baselines such as Minimax, D4M, IGD, and MGD3, etc., which represent current frontiers of DD and they also addressed the same issue of diversity distillation as PRISM.

**Questions:**

- **Confounding from powerful teachers**: The paper does not disentangle the influence of teacher networks from the PRISM framework itself. It remains unclear whether the observed gains stem from the architectural decoupling or simply from using the more powerful teachers itself. Can you provide the ablation results with different single BN teachers?
- **Diversity metrics**: Can you report diversity metrics relative to the original dataset to clarify whether PRISM preserves or exceeds raw data variation? If the behavior is different, how to explain the performance gains?
- **Confused fig1**: I don't believe the classification accuracy of the fig1(right) can be better than the left ones since the decision boundary is much more mixed. If I missed something, please let me know.
- **Experiments**: Would the benefits of PRISM persist on lower-resolution or domain-shifted dataset?
- **Incremental contribution**: The use of multiple teacher models closely resembles the model pool concept in prior dataset distillation methods. The paper should make it clearer what **fundamentally** (rather than just framework) differentiates PRISM from these existing frameworks.

---

### Official Review · Reviewer_LBFu · 2025-10-31

**Soundness:** 3
**Presentation:** 2
**Contribution:** 2
**Rating:** 2
**Confidence:** 3

**Summary:**

- This paper presents PRISM, a dataset distillation algorithm which uses an ensemble of teacher models to generate distilled images
- The main idea is to take the SRe2L loss, and rotate through a pool of teacher models for the batchnorm loss, while using the same model for label alignment loss
- Empirical results show good improvement over baselines

**Strengths:**

- Pretty intuitive motivation, and presented clearly
- Strong empirical results
- Thorough ablation studies

**Weaknesses:**

- Relatively incremental contribution. The use of multiple architecture backbones has been studied in much prior work [1, 2], along with the choice of using teacher emsembles for prediction generation. In particular [2], uses a very similar idea of rotating teacher models during optimization, albeit using an ensemble of the same resnet-18 model retrained multiple times (rather than different backbones) in this paper

- The marginal contribution of this work over previous work seems minor. In particular, from Table 1, gains are rather small over the baselines presented. Furthermore, PRISM, as discussed in section 4.4, uses pretty much every optimization trick presented in all the prior work, so it is unclear what the exact contribution is. For example, the margin between EDC and PRISM in table 1 is rather small, but PRISM additionally uses the "varying the distillation iterations per image" technique from DELT. So it is unclear, for example, if EDC with that trick would outperform PRISM (given than the gains from table 3b appear to be ~1.3%, approximately the same as the margin between PRISM and EDC).

- Privacy analysis in Appendix is completely incorrect. Differential privacy can only be guaranteed with two main components: **bounded sensitivity**, and **calibrated noise addition**. PATE adds privacy by adding calibrated noise to the queries of the teacher. Furthermore, this privacy loss is composed over every query to these teacher models. PRISM claims "The ”noise” required for privacy is not explicitly added but arises organically from the inherent disagreement between the gradients of two distinct model architectures", however without clipping gradients from these teachers, or calibrated the actual noise scale, there is no privacy bound whatsoever, so it is useless to discuss differential privacy. Furthermore, teachers in PRISM are trained on the full dataset, while in PATE, they are trained on subsets which is necessary for privacy. I strongly suggested removing this discussion or adding a strong disclaimer that there are **no real privacy benefits** afforded by using multiple teachers, and everything in that section is purely speculative.



[1] Generalized Large-Scale Data Condensation via Various Backbone and Statistical Matching

[2] Large Scale Dataset Distillation with Domain Shift

**Questions:**

- What are the results in terms of cross-architecture generalization? Given that the method uses multiple backbones, we would expect better architecture generalization.

---

### Note · Authors · 2025-11-13

I have read and agree with the venue's withdrawal policy on behalf of myself and my co-authors.